# Application of Fractal Theory to the Analysis of Failure Characteristics of Low-Velocity-Impact Concrete Slabs

**Song Gu** [1], **Jiachen Zhao** [1,*], **Jinxing Li** [2], **Feng Peng** [3], **Chao Kong** [1] and **Liqiong Yang** [1]

[1]  School of Civil Engineering and Architecture, Southwest University of Science and Technology, Mianyang 621010, China; gusong@swust.edu.cn (S.G.); kongchaokc@foxmail.com (C.K.); ylq@ustc.edu.cn (L.Y.)
[2]  Sichuan Bangtai Investment Group Co., Ltd., Chengdu 610041, China; 17623858789@163.com
[3]  College of Environment and Civil Engineering, Chengdu University of Technology, Chengdu 610059, China; pengfeng@stu.cdut.cn
*   Correspondence: zzzjiachen@163.com

**Abstract:** The fractal characteristics of low-velocity-impact concrete slabs were studied using fractal theory, and the fractal dimension value of cracks of each concrete specimen plate was calculated using box dimension as the basic principle and digital image analysis technology in conjunction with MATLAB software (R2021b) calculation functions. The energy dissipation of concrete slabs during low-velocity impact is calculated using the elastic sheet theory. The calculation results are realistic, and the energy conversion of concrete slabs during low-velocity impact is analyzed based on these results. The research findings indicate that concrete slab cracks exhibit good fractal characteristics during low-velocity impact, and their values can be utilized as a parameter to determine the extent of concrete slab failure. Moreover, the study found that the fractal dimension value of concrete slab cracks and the associated plastic deformation energy display good exponential function characteristics in the energy dissipation mechanism of low-velocity concrete slabs.

**Keywords:** fractal theory; concrete slab; impact; crack; energy

## 1. Introduction

As the most widely used engineering material in the field of industrial, civil, and military protection engineering, the mechanical properties, especially the dynamic mechanical properties, of concrete have always been the focus of researchers' attention [1]. Concrete structures are often subjected to various natural or manmade explosions, impacts, and other dynamic loads during service, causing irreparable damage to the structure and making it lose its normal service function [2]. The study of dynamic response and failure characteristics of concrete structures under impact is one of the important topics in the field of civil engineering disaster prevention and mitigation [3].

Since B.B. Mandelbrot et al. [4] first applied fractal theory to the quantitative analysis of fracture surface characteristics of metallic materials in 1984, fractal theory has been developed and applied to the study of concrete materials and structures. Kirong Zhou et al. [5] used fractal geometry theory to study the dimensional effect law of concrete axial compressive strength, and they established the quantitative expression of the dimensional effect law with fractal dimension. A. Carpinteri and M. B. Bruneto [6–8] used fractal theory to study the size effect in concrete and rock fractures; Guangzong Kang et al. [9] studied the fractal behavior of the crack width size effect of concrete structures and confirmed that the crack width of concrete structures has an obvious dimensional dependence. Yebing Sun et al. [10] used a fractal and fracture damage mechanics theory to quantitatively analyze and describe damage cracking and cracks' growth paths in reinforced concrete beams. Qian Wang et al. [11] used fractal theory to study the development law of surface cracks of reinforced concrete beams and explored the possibility of applying fractal theory to

evaluate the degree of damage and durability of reinforced concrete components. Chen et al. [12] studied the crack growth in concrete at different strain rates and found that the development and failure of multiple cracks in concrete aggregate at low strain rates had a significant impact on its dynamic performance. However, with an increase in strain rate, the influence of the inertial effect on its dynamic performance becomes dominant. Mei et al. [13] applied the crushing fractal theory to study the stress resistance of rigid projectile impact concrete targets and derived a calculation formula for quasi-static stress resistance. The reviewed literature suggests that fractal theory has application in the study of static and dynamic mechanics of concrete materials or structures.

Simultaneously, several studies evaluating concrete structure failure characteristics under the influence of impact have been conducted. In one such study, Shuang Wang [14] employed finite element software LS-DYNA to simulate rockfall impact on shed frameworks. The study reveals that both impact depth and impact force increase with energy and the damage cover area assumes an "X" shape, which expands correspondingly to energy. K Senthi et al. [15] subjected reinforced concrete slabs to low-speed impact loads and observed punch failures in all failed specimens. They proposed an empirical formulation to appraise the bearing capacity of reinforced concrete slabs under low-speed impact. Dongpo Wang [16], in contrast to other traditional reinforced concrete slabs founded on empirical or Hertzian contact theory, stresses the difference from current circumstances. Therefore, the paper suggests incorporating the Olson theory's control equation of orthotropic composite plate to understand the impact dynamics. Subsequently, it recommends performing a dynamic response investigation of elastoplastic contact theory under rolling stone impact on reinforced concrete slabs. Zineddin [17] conducted impact experiments on concrete slabs at various speeds. The findings revealed that different failure modes occur at different impact speeds, including bending, shearing, and punching. Additionally, concrete spalling and crushing generally occur at the bending failure stage. Yongjun Deng [18] and colleagues conducted both target tests and numerical simulations to investigate projectile penetration in reinforced concrete. The study found that the concrete pressure was highest near the projectile's trajectory, and the stress performance of the steel bar differed in various zones. Under low-velocity impact, the concrete structure experiences a significant load change in a short time. As a result, the concrete structure tends to crack along the direction of its plastic hinge. High-velocity impact, also known as high-velocity penetration, results in greater instantaneous load changes, causing more severe damage to the concrete structure. Under high-velocity impact, the concrete structure is likely to suffer from spalling, failure, and scab cracks.

Related to the impact of concrete, scholars from domestic and foreign regions have conducted various studies, primarily concentrating on analyzing the fractal features of concrete's dynamic response and failure. The problem of high-velocity impact is mainly researched in military and high-level protection engineering, where the research done is relatively comprehensive. This paper utilizes the damage effect test conducted on a low-velocity-impact concrete slab [19] to simulate the protective concrete slab structure that could be affected by rockfall. Analyzed through fractal theory and energy calculation, this paper explores the energy transformation process of concrete slabs under low-velocity impact. This approach suggests a new research direction for studying the damage mechanism of the concrete slab's structure under an impact. Moreover, these findings can serve as a guide in the design of relevant civil protection engineering.

## 2. Experiment

### 2.1. Experimental Design

Three variables (thickness, impact velocity (fall height), and concrete strength) are mainly considered in the test, and the specimens are 500 mm × 500 mm thick, smooth concrete slabs with thicknesses of 40 mm and 80 mm (due to the thin thickness of the precast concrete slab in the test, a separate strength test was conducted for each specimen to ensure the accuracy of the test). The test takes into account only the most adverse

working conditions of impact [20], that is, when the drop weight is perpendicular to the plane of the concrete slab and the sample plate is fixed to the test bench in the form of four simple supports.

In order to facilitate the observation of the failure of the low-velocity-impact concrete slabs, a set of homemade simple drop-weight impact devices was adopted, as shown in Figure 1, which mainly consists of fixed pulley, wire rope, drop weight, hemispherical impact head, and impact force sensor.

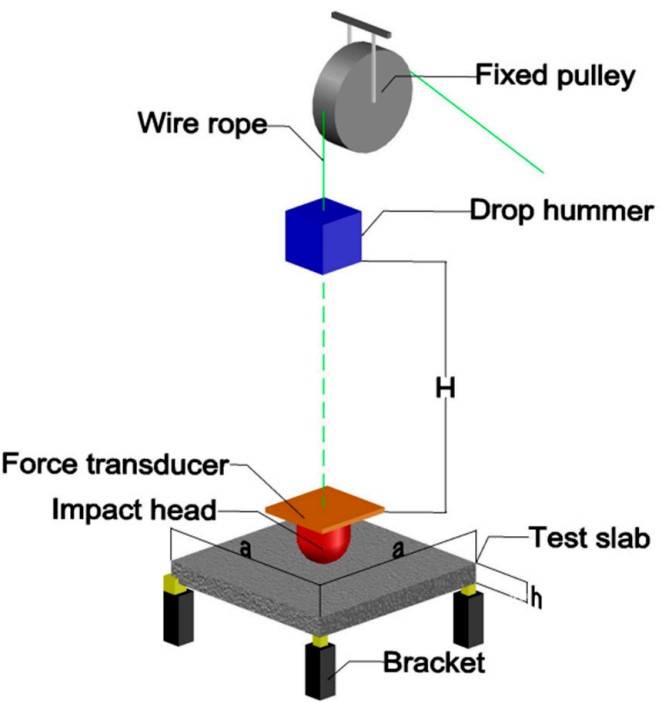

**Figure 1.** Diagram of test device.

*2.2. Experiment Results*

The experiment results are summarized in Table 1.

**Table 1.** Summary of the experiment results.

| Specimen number | h4-05-1 | h4-05-2 | h4-1-1 | h4-1-2 | h4-1-3 | h8-05-1 | h8-05-2 | h8-1-1 | h8-1-2 | h8-1-3 | h8-2-1 | h8-2-2 | h8-3-1 | h8-3-2 |
|---|---|---|---|---|---|---|---|---|---|---|---|---|---|---|
| Strength Mpa | 17.6 | 10.1 | 10.3 | 7.6 | 22.3 | 27.4 | 33.1 | 23.9 | 30.1 | 23.6 | 31.8 | 25.8 | 20.6 | 31.2 |
| Velocity m/s | 3.16 | 3.16 | 4.47 | 4.47 | 4.47 | 3.16 | 3.16 | 4.47 | 4.47 | 4.47 | 6.32 | 6.32 | 7.74 | 7.74 |
| Peak impact force kN | 40.1 | 33.62 | 33.82 | 32.58 | 37.71 | 47.67 | 43.98 | 55.72 | 64.46 | 62.12 | 83.91 | 80.17 | 122.44 | 115 |

(Note: h4-1-1 means thickness 40 mm, drop weight height 1 m (05 means 0.5 m); 1st sample).

The failure patterns of each specimen are shown in Figure 2 below. For more information on the test of rockfall low-velocity-impact concrete slabs, see Literature [19].

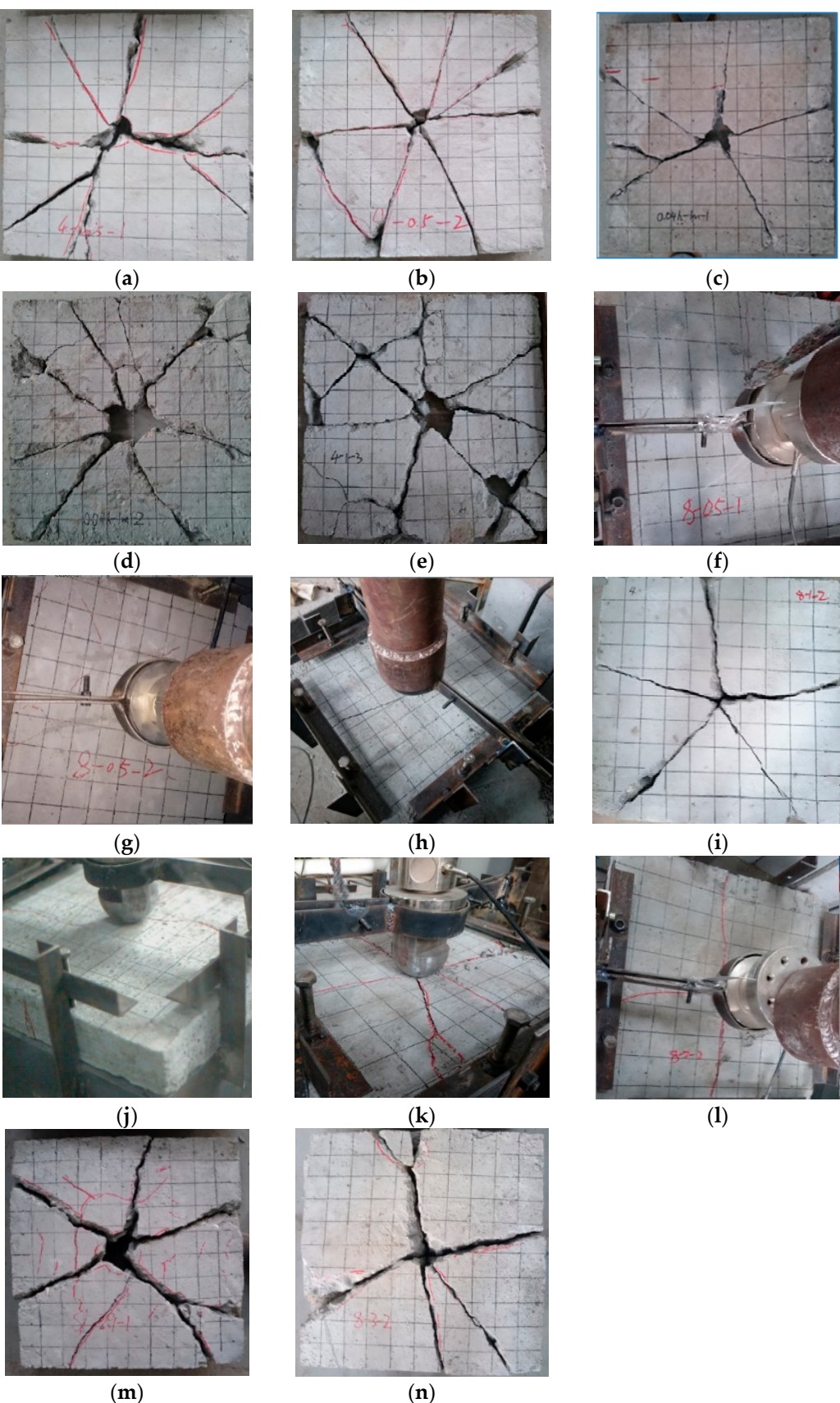

**Figure 2.** Diagram of the failure pattern of the specimen: (**a**) h4-05-1; (**b**) h4-05-2; (**c**) h4-1-1; (**d**) h4-1-2; (**e**) h4-1-3; (**f**) h8-05-1; (**g**) h8-05-2; (**h**) h8-1-1; (**i**) h8-1-2; (**j**) h8-1-3; (**k**) h8-2-1; (**l**) h8-2-2; (**m**) h8-3-1; (**n**) h8-3-2.

### 3. Fractal Calculation of Cracks in Concrete Slabs

Fractal geometry is a new branch of mathematics introduced by the French mathematician Benoit B. Mandelbrot. Unlike traditional geometry (e.g., Euclidean geometry, differential geometry), fractal geometry can describe, study, and analyze at a deeper level those chaotic, irregular, and random natural phenomena that are common in nature [21].

#### 3.1. Box Dimension Definition

For irregular images, not all have fractal characteristics, which produces a quantitative definition to describe the process of structural fractal phenomenon characteristics, that is, fractal dimension *D*. Fractal dimension is an important parameter of fractals that quantitatively represents the essential characteristics of fractal sets, and the solution of its values is the key to solving practical problems with fractal concepts. The fractal dimension is commonly referred to as the box dimension, while the calculation method of the fractal dimension of cracks on the concrete surface is the box-counting method [22].

The basic principle of calculating the fractal dimension of cracks on the surface of concrete slabs under impact is to use square grids (shaped like boxes) of different scales, r, to cover cracks on the surface of concrete slabs (as shown in Figure 3) and to calculate the total number of boxes covering surface cracks, *N*(r), for boxes of a given scale, r [23]. Assuming that step *i* uses a box cover of scale *i*, the total number of boxes required is $N_i(i)$, and step *i* + 1 is covered with a box of scale *i* + 1, and the total number of boxes required is $N_{i+1}(i + 1)$, then the ratio of the number of boxes required at any two scales to the scale can be described by Equation (1):

$$N_{i+1}/N_i = (\delta_i/\delta_{i+1})^D \qquad (1)$$

where the variables *i* and *i* + 1 represent the number of steps to count. The symbol '$\delta_i$, $\delta_{i+1}$' denotes the size of the box when calculating in steps *i* and *i* + 1. Additionally, the symbol '$N_i$, $N_{i+1}$' describes the size at steps *i* and *i* + 1. The total number of boxes needed to cover cracks on the surface of the concrete slab is represented by n. *D* represents the fractal dimension of the cracks on the surface of the concrete slab.

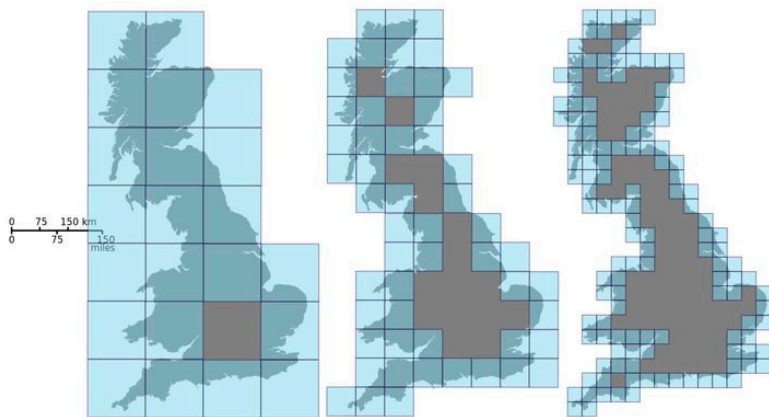

**Figure 3.** Schematic diagram of box-counting method.

#### 3.2. The Principle of Box Dimension Calculation

According to the definition of box dimension, the fractal dimension value of a concrete slab crack under impact is calculated by the digital image binarized information matrix and MATLAB software, and the calculation steps are as follows:

(1) Import the image into Matlab calculation software, and if the image size is not 2 n × 2 n pixels, binate it to generate a binary matrix of the image (the matrix elements are only 0 or 1);

(2)   Start from 20 × 20 minimum boxes and scan all submatrices to get the total number of boxes needed;

(3)   Change the box size to get the total number of boxes under different sizes until the box size is equal to the image size;

(4)   Take the logarithm of the obtained box size and the total number of boxes and draw the fitted curve. The slope of the fitted curve is the fractal dimension value of the image.

### 3.3. Box Dimension Calculation

A crack pattern of the concrete slab specimen after impact was arbitrarily selected for calculation and demonstration, and the crack pattern of the selected specimen slab is shown in Figure 4a. The surface crack pattern of the concrete slab was imported into MATLAB calculation software, and the black and white bitmap was first obtained by binary processing, and then the box dimension value was obtained by the internal plug-in Fraclab of MATLAB software, $D_{h8-1-2} = 1.33$, and the binary processing diagram and fitting results were intercepted, as shown in Figure 4.

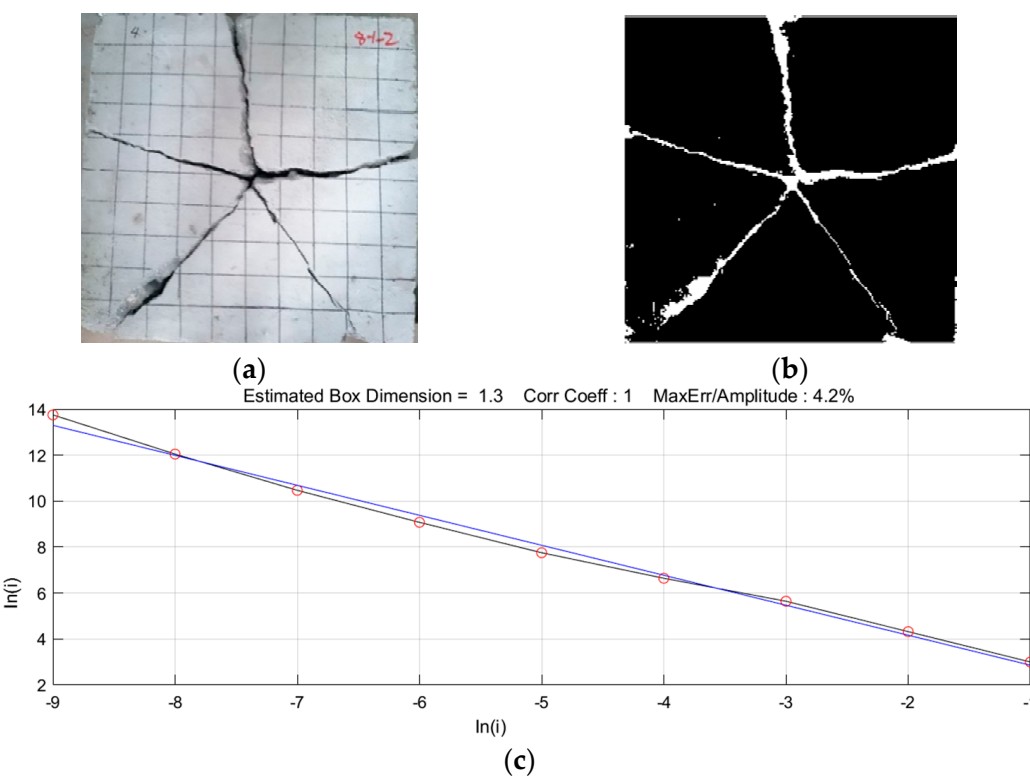

**Figure 4.** Calculation process of fractal dimension *D* of fracture in specimen plate: (**a**) image of cracks in the specimen plate; (**b**) binarization of images; (**c**) fractal dimension value *D*.

## 4. Energy Calculation of Concrete Slabs

The failure and deformation process of a concrete slab under the action of low-speed impact can be roughly divided into three stages: elastic deformation stage, elastoplastic deformation stage, and plastic deformation to failure stage. The results show that when the concrete slab is damaged by low-velocity impact, the total energy, $W_T$, absorbed by the specimen is mainly composed of three parts: (1) elastic deformation energy, $W_E$; (2) plastic deformation energy, $W_F$, of concrete slab specimens; (3) energy, $W_0$, consumed in the form of heat energy, sound wave energy, etc. in the process of influencing. According to the law of conservation of energy, we obtain Equation (2):

$$W_T = m_{\text{Drop weight}}gh = W_E + W_F + W_0 \tag{2}$$

where $m_{\text{Drop weight}}$ is 40.55 kg, $g$ takes 9.81 m/s$^2$, and $h$ is the drop weight release height.

The total energy absorbed during the fracture of the specimen is mainly used for the fracture energy, damage energy, and kinetic energy of the debris spray [24]. Therefore, it is assumed that the plastic deformation energy, $W_F$, of the concrete slab specimen is composed of the initial flaw propagation of the concrete slab specimen, the new cracks, and the energy dissipation of the specimen splinter fragments upon impact. It is generally believed that when the loading rate is not particularly high, the energy consumed, $W_0$, in the form of heat energy, sound wave energy, etc. is small and can be ignored [25]. Equation (2) can be simplified and expressed as Equation (3):

$$W_T = W_E + W_F \tag{3}$$

*Elastic Deformation Energy of Concrete Slabs*

In order to calculate the elastic deformation energy of concrete slabs, this paper calculates the maximum elastic deformation of concrete slabs based on the small deflection bending problem of thin slabs in the elastic mechanics theory and calculates the deformation energy required for the maximum elastic deformation of concrete slabs by using the broad definition of strain energy.

In elasticity, an object bounded by two parallel planes and a cylinder perpendicular to these two parallel planes is called a plate. The distance between two plate surfaces, δ, is called the thickness of the plate, while the plane with δ bisecting the thickness is called the center surface of the plate. If the thickness of the plate, δ, is much smaller than the minimum size, b, of the center surface (e.g., less than b/8 to b/5), it is called a thin plate [26]. The following basic assumptions are made about the deformation and stress of thin plates, often referred to as the Kirchhoff–Love hypothesis.

(1)  The points on the mid-surface of the plate do not produce displacements parallel to the mid-surface;
(2)  The normal perpendicular to the mid-surface is still a straight line after deformation and perpendicular to the mid-surface after deformation.

The concrete slab studied in this paper meets the definition of a thin plate structure, where the middle surface of the concrete slab is taken as the XY plane, and the Z axis and the XY axis are spiraled by the right hand and perpendicular to the middle plane to establish a coordinate system, as shown in Figure 5.

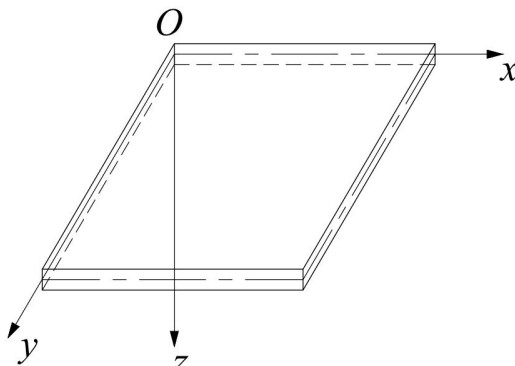

**Figure 5.** Sheet coordinate system.

From the thin plate theory of elasticity, the bending differential equation represented by the midplane deflection, $w(x,y)$, is expressed as Equation (4):

$$\frac{\partial^2 M_x}{\partial x^2} + 2\frac{\partial^2 M_{xy}}{\partial x \partial y} + \frac{\partial^2 M_y}{\partial y^2} = \frac{q}{D} \tag{4}$$

The concrete slab is simplified to a square plate with a simple side length around it, and the elastic deformation problem is subjected to a concentrated force, *P*, at the center of the slab, as shown in Figure 6.

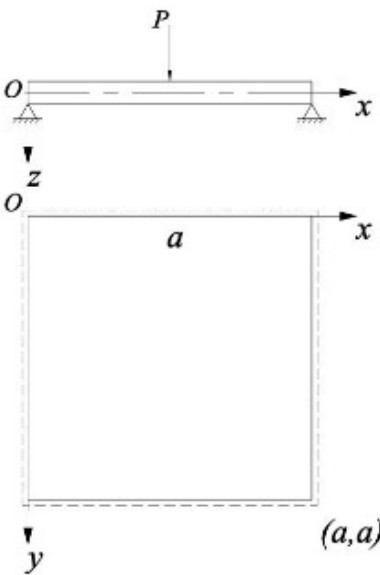

**Figure 6.** Simplified mechanical model.

Then we expand the bending function, *w(x,y)*, and the load, *q(x,y)*, into a double trigonometric series and satisfy the boundary condition given by the quadrilateral, assuming that the solution of Equation (4) is Equation (5):

$$w(x,y) = \sum_{m=1}^{\infty} \sum_{n=1}^{\infty} A_{mn} \sin\frac{m\pi x}{a} \sin\frac{n\pi y}{a} \tag{5}$$

where *m* and *n* are arbitrary positive integers, and $A_{mn}$ is the coefficient.

By replacing Equation (4) with Equation (5), we obtain Equation (6).

$$\pi^4 D \sum_{m=1}^{\infty} \sum_{n=1}^{\infty} \left(\frac{m^2}{a^2} + \frac{n^2}{a^2}\right) A_{mn} \sin\frac{m\pi x}{a} \sin\frac{n\pi y}{a} = q(x,y) \tag{6}$$

where *D* is the bending stiffness of the plate, calculated by Equation (7).

$$D = \frac{E\delta^3}{12(1-\nu^2)} \tag{7}$$

where *E* is the modulus of elasticity of concrete, $\delta$ is the thickness of concrete slab, $\nu$ is the Poisson's ratio of concrete, and the general value is 0.2.

Simultaneously, *q(x,y)* on the right-hand side of Equation (6) transforms into a double triangle series, resulting in Equation (8).

$$\pi^4 D \sum_{m=1}^{\infty} \sum_{n=1}^{\infty} \left(\frac{m^2}{a^2} + \frac{n^2}{a^2}\right) A_{mn} \sin\frac{m\pi x}{a} \sin\frac{n\pi y}{a} = \sum_{m=1}^{\infty} \sum_{n=1}^{\infty} C_{mn} \sin\frac{m\pi x}{a} \sin\frac{n\pi y}{a} \tag{8}$$

Using the orthogonality of the trigonometric function, it is finally possible to obtain the equation of flexure of the four-sided simple square plate under a concentrated force load in the center of the plate, that is, Equation (9).

$$w = \frac{4P}{\pi^4 a^2 D} \sum_{m=1,3,5\cdots}^{\infty} \sum_{n=1,3,5\cdots}^{\infty} (-1)^{\frac{m+n}{2}-1} \frac{\sin\frac{m\pi x}{a} \sin\frac{n\pi y}{a}}{\left(\frac{m^2}{a^2} + \frac{n^2}{a^2}\right)^2} \tag{9}$$

As each concrete slab belongs to self-made specimens, its concrete strength varies. The "Code for the Design of Concrete Structures" [27] states that the modulus of elasticity differs among concrete of different strengths. To ensure accurate calculations, it is necessary to substitute the concrete strength and modulus of elasticity, E, for each specimen slab into Equation (7) to determine the genuine flexural stiffness of each concrete slab specimen.

$V_\varepsilon$ can express the energy stored in the member due to elastic deformation. The broad definition of Equation (10) allows for the calculation of strain energy, $V_\varepsilon$.

$$V_\varepsilon = \frac{1}{2}F\Delta \tag{10}$$

where $F$ is a generalized force that can represent a force, a couple, or a combination of both. $\Delta$ is a generalized displacement and can represent a line displacement, an angle displacement, or a combination of both.

The elastic deformation of the concrete slab under impact comprises mainly the lower deflection and the angular displacement that accompanies it. The elastic deformation energy of the slab can be obtained by using Equation (11) in combination with the generalized strain energy Equation (10).

$$W_E = \iiint_V P \cdot w + M_P \theta dV = \iiint_V P \cdot w + P\sqrt{(250-x)^2 + (250-y)^2} \cdot \frac{w}{\sqrt{x^2+y^2}} dV \tag{11}$$

Note: This formula is based on the coordinate system shown in Figure 6, where the origin, O, is located at the upper left corner of the square plate. The variable 'Point' represents the distance of any point in the system from the center point, which is located at (250, 250).

The elastic deformation energy calculation program of the concrete slab uses Equation (10) and is programmed by MATLAB calculation software. The maximum impact force, determined through the impact head force sensor, is used to calculate the size of the elastic deformation energy, $W_E$, under the action of low-speed impact. The program calculation flow chart is shown in Figure 7.

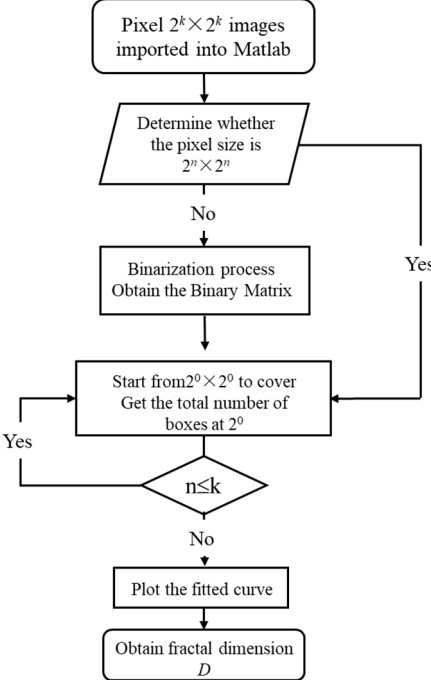

**Figure 7.** Flowchart calculation.

The data from the drop-weight impact test is collected, and the maximum impact force of each specimen plate is recorded in the calculation program to obtain the elastic deformation energy, $W_E$, value of each plate. The resulting values are presented in Table 2.

**Table 2.** The result of elastic deformation can be calculated.

| Specimen Number | $W_E$/J | Specimen Number | $W_E$/J | Specimen Number | $W_E$/J | Specimen Number | $W_E$/J | Specimen Number | $W_E$/J | Specimen Number | $W_E$/J |
|---|---|---|---|---|---|---|---|---|---|---|---|
| h4-05-1 | 3.391 | h4-1-1 | 2.577 | h8-05-1 | 0.943 | h8-1-1 | 1.414 | h8-2-1 | 2.724 | h8-3-1 | 6.827 |
| h4-05-2 | 2.991 | h4-1-2 | 2.810 | h8-05-2 | 0.748 | h8-1-2 | 1.608 | h8-2-2 | 2.667 | h8-3-2 | 5.117 |
| | | h4-1-3 | 3.028 | | | h8-1-3 | 1.757 | | | | |

## 5. Fractal Analysis of Concrete Slabs

### 5.1. Fractal Analysis of Failure Characteristics

The fractal dimension of each concrete specimen's crack was obtained via digital image analysis technology and MATLAB calculation function and is presented in Table 3.

**Table 3.** Fractal dimensions of fractures in specimen plates.

| Specimen Number | Fractal Dimension | Maximum Error/% | Specimen Number | Fractal Dimension | Maximum Error/% |
|---|---|---|---|---|---|
| h4-05-1 | 1.56 | 2% | h8-1-1 | 1.26 | 2.8% |
| h4-05-2 | 1.52 | 2.8% | h8-1-2 | 1.3 | 4.2% |
| h4-1-1 | 1.25 | 2.2% | h8-1-3 | 1.17 | 9.7% |
| h4-1-2 | 1.56 | 1.1% | h8-2-1 | 1.26 | 4% |
| h4-1-3 | 1.51 | 1.7% | h8-2-2 | 1.18 | 5.6% |
| h8-05-1 | 1.01 | 15% | h8-3-1 | 1.47 | 2.3% |
| h8-05-2 | 0 | - | h8-3-2 | 1.37 | 3.5% |

According to the box dimension calculation theory, the length, width, and hole area of cracks formed in concrete slabs due to impact are related to the number of "boxes" required to cover them at different scales. Consequently, the wider the crack length and width (or the larger the hole area), the larger the calculated fractal dimension. Cracks (even holes) in concrete slabs due to impact are an indication of their degree of failure. The fractal dimension of these cracks can serve as a reference index for the degree of failure.

The failure pattern of each concrete slab specimen is that the crack penetrates radially to the four corners with the impact point as the center, and holes are formed at the impact point position, which indicates the performance of the slab against bending under the action of impact. However, some concrete slab specimens have tapered holes at the impact center, indicating that the concrete slab also has some punching failure characteristics under the impact. Combined with the failure pattern diagram of the specimen in Figure 1, it can be seen that the more severe the failure of the specimen, the larger the fractal dimension value.

### 5.2. Fractal Analysis of Impact Energy

This section investigates the correlation between impact energy and fractal dimensions of self-made concrete slabs. It should be noted that inconsistent concrete strength may exist among the test samples. The concrete strengths of h8-05-1, h8-1-2, h8-2-1, and h8-3-2 are uniform, except for specimen h8-05-1, which has a strength of 25 MPa instead of 30 MPa. Based on the experimental control principle, the only variable in this specimen set is the rockfall release height, i.e., the impact energy. Therefore, specimen sets h8-05-1, h8-1-2, h8-2-1, and h8-3-2 were chosen as the research subjects.

The impact energy of each specimen in the selected study object is calculated using Equation (2). The fractal dimension values from Table 4 are then combined into a new table (also shown in Table 4) and used to plot the trend of the impact energy of concrete slabs along with its corresponding fractal dimension values (illustrated in Figure 8).

**Table 4.** Object plate crack impact energy and fractal dimension.

| Drop Hammer Release Height/m | Impact Energy/J | Fractal Dimension | Max. Error/% |
|---|---|---|---|
| 0.5 | 198.89 | 1.01 | 15 |
| 1.0 | 397.79 | 1.3 | 4.2 |
| 2.0 | 795.91 | 1.26 | 4 |
| 3.0 | 1193.38 | 1.37 | 3.5 |

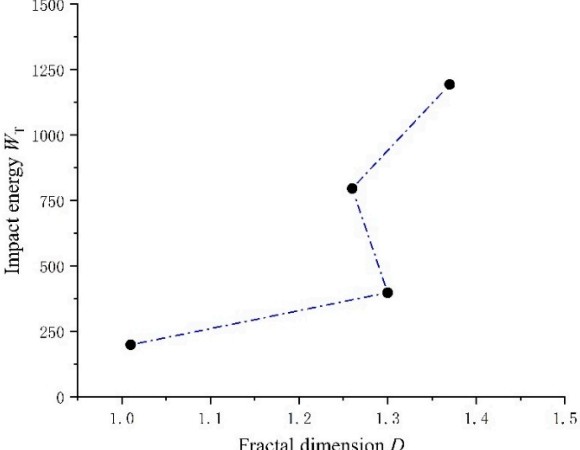

**Figure 8.** Trend graph of the impact energy of the object plate and its fractal dimension.

Assuming that the thickness of the specimen plate is constant, increasing the height from which the drop weight is released results in a higher impact energy of the concrete plate and an increasing trend in its fractal dimension.

The data received from the force sensor in the low-speed impact test is collected and processed to obtain the impact force time history curve for the test pieces h8-05-1, h8-1-2, h8-2-1, and h8-3-2. Figure 9 shows the impact force time history curve. The whole impact process's duration was 0.07 s, indicating that the impact energy had no significant effect on the wave's duration. Additionally, in each test piece's impact time history curve, there were two primary impact force peaks. When the impact head meets the concrete specimen plate, it starts to vibrate due to the impact dynamic load. Then, the impact force instantly reaches the first peak after a short fluctuation lasting about 0.01 s. Following the second fluctuation, the impact force reaches the second peak. After this point, the impact energy dissipates, and the impact force gradually weakens until it returns to zero.

### 5.3. Fractal Analysis of Plastic Deformation Energy

The tested slab is constructed from plain concrete that exhibits brittle characteristics. Upon exceeding its elastic deformation limit, the concrete slab undergoes plastic deformation that is manifested mainly as overall bending of the slab and the formation and propagation of cracks.

The study's accuracy may be affected since the concrete slab is a self-fitting specimen, and the test results have some discreteness. Therefore, h8-05-1, h8-1-2, h8-2-1, and h8-3-2 were selected as research objects to examine the relationship between the plastic deformation energy of the concrete slab and its fractal dimension. Equation (3) provided the plastic deformation energy of the specimen, and the calculation results are showcased in Tables 2 and 5. Figure 10 displays the plot of the fractal dimension, *D*, of the cracks of the chosen specimen and the corresponding plastic deformation energy trend.

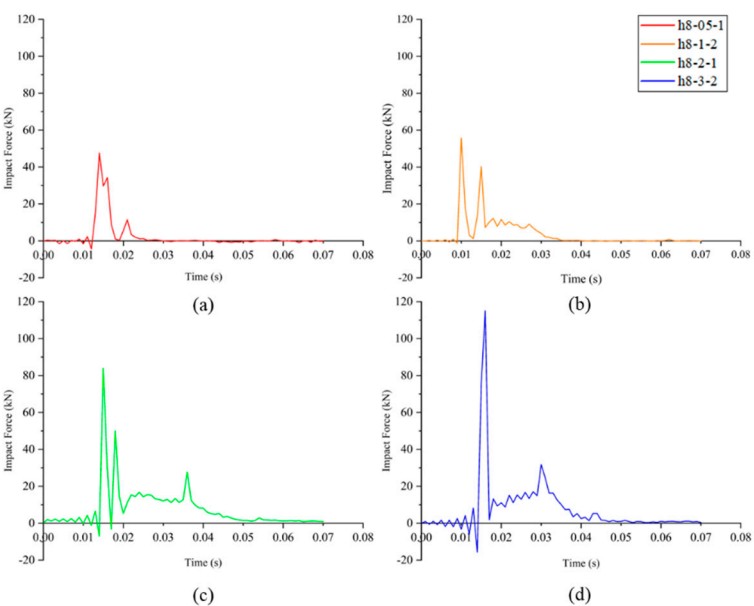

**Figure 9.** The time-history curve of the impact force on the test specimen slabs: (**a**) h8-05-1, (**b**) h8-1-2, (**c**) h8-2-1, (**d**) h8-3-2.

**Table 5.** Plastic deformation energy of test pieces.

| Specimen Number | $W_T$ (J) | $W_E$ (J) | $W_F$ (J) | $D$ |
|---|---|---|---|---|
| h8-05-1 | 198.897 | 0.748 | 198.149 | 1.01 |
| h8-1-2 | 397.795 | 1.414 | 396.381 | 1.3 |
| h8-2-1 | 795.591 | 2.667 | 792.924 | 1.26 |
| h8-3-2 | 1193.387 | 5.117 | 1188.27 | 1.37 |

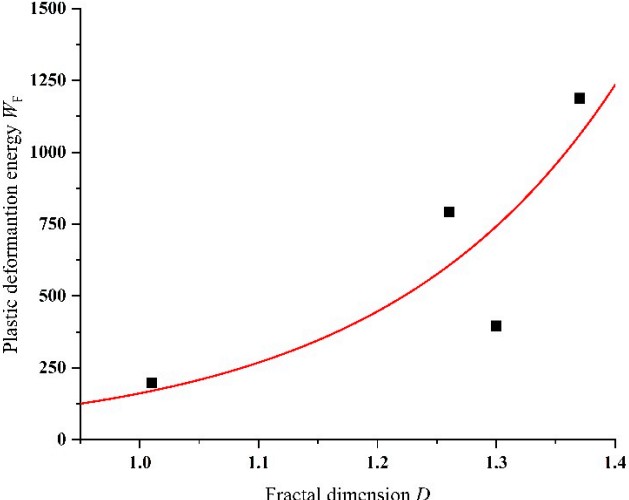

**Figure 10.** Trend chart of plastic deformation energy and fractal dimension of concrete slab.

The Figure 10 above illustrates an increase in plastic deformation energy, $W_F$, of the concrete slab with an increase in fractal dimension, $D$. The analysis shows a strong exponential function relationship between the two, represented in Equation (12) as

$$W_F = 161.51^D \tag{12}$$

where the Equation (12) has a coefficient of determination (COD) of 0.70. The COD value ranges from 0 to 1, with a higher value indicating a better fitting effect.

Due to the limited number of specimens and the discreteness of the test results, the quantitative relationship between the plastic deformation energy of the concrete slab and the fractal dimension of the crack should be used with caution in actual engineering situations [28]. However, this formula can provide a preliminary estimate of the energy consumption under certain circumstances by using the value of the fractal dimension of the concrete slab crack under impact. Calculation of the plastic deformation energy of the concrete specimen plate demonstrates that the impact energy of a concrete slab under low-velocity impact is primarily converted into plastic deformation energy, representing the primary energy dissipation mechanism of the concrete.

## 6. Conclusions

This paper examines the fractal analysis of the failure effect of concrete slabs under impact given the shortcomings of the existing structure when subjected to impact. This analysis is based on the failure test of low-velocity-impact concrete slabs and draws the following conclusions:

(1) The fractal dimension of low-velocity-impact concrete slabs' cracks was calculated using digital image analysis technology and MATLAB calculation function. This revealed that the cracks in concrete structural members exhibit good fractal characteristics under impact.

(2) During low-velocity impact, the fractal dimension, *D,* of cracks on the surface of concrete slabs can be utilized to characterize the degree of failure of those slabs. A higher fractal dimension of the crack in the concrete slab indicates a more severe degree of damage.

(3) Impact energy of the concrete slab increased together with the drop weight release height. The corresponding fractal dimension values additionally demonstrated an upward trend, indicating a positive correlation between the two.

(4) By means of theoretical calculation and MATLAB calculation software, we designed a plastic deformation energy calculation program for concrete slab under low-velocity impact. Combined with the numerical fractal dimension analysis of cracks, it was discovered that the two exhibit a good exponential function relationship; this could provide promising research opportunities for revealing the failure mechanism of concrete slabs under low-velocity impact.

**Author Contributions:** S.G.: data curation, formal analysis, investigation, methodology, software, validation, and writing—original draft. J.Z.: methodology and writing—original draft. J.L.: data curation, validation, and writing—original draft. F.P.: validation and writing—original draft. C.K.: methodology and validation. L.Y.: methodology and writing—review and editing. All authors have read and agreed to the published version of the manuscript.

**Funding:** This research was funded by the National Natural Science Foundation of China, grant number 52108385.

**Data Availability Statement:** Not applicable.

**Acknowledgments:** The authors gratefully acknowledge the support provided by the National Natural Science Foundation of China (Grant No. 52108385). The authors also would like to express their gratitude to the reviewers for their checks and approvements.

**Conflicts of Interest:** All authors of this manuscript have directly participated in the planning, execution, and analysis of this study. The contents of this manuscript have not been copyrighted for publication previously. The contents of this manuscript are not now under consideration for publication elsewhere. The contents of this manuscript will not be copyrighted, submitted, or published elsewhere while accepted in *Buildings*. There are no directly related manuscripts or abstracts, published or unpublished, by any of the authors of this manuscript. We declare that, excepting "the National Natural Science Foundation of China", we have no financial or personal relationships with other people or organizations that can inappropriately influence our work, and there is no professional or other personal interest of any nature or kind in any product, service, and

company that could be construed as influencing the position presented in, or the review of, the manuscript entitled.

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
