# Peer review of "Application of Fractal Theory to the Analysis of Failure Characteristics of Low-Velocity-Impact Concrete Slabs"

_buildings, doi:10.3390/buildings13092190_

Round 1

Reviewer 1 Report

This article studies the fractal characteristics of low-speed impact concrete slabs. Using digital image analysis technology and MATLAB software calculation function, the fractal dimension values of each sample crack are calculated. Based on this, the energy conversion during low-speed impact is analyzed to study the impact failure mechanism of concrete slab structures, which has a certain significance. However, there are still some shortcomings as follows:

1. It is recommended to use reasonable expressions, such as formula (1), to explain the meaning of the parameters in the formula.

2. When exploring the relationship between impact energy and fractal dimension of concrete slabs, an explanation should be given as to why h8-05-1, h8-1-1, h8-2-1, and h8-3-2 are chosen.

3. In Figure 8, when the impact energy is 198.89, the fractal dimension does not correspond to the values in Table 4 and should be carefully checked and verified.

4. Formula (12), the parameter explanation given below is COD=0.70. What parameter is COD? It is not included in the formula and should be carefully checked for writing errors.

5. The details in the article should be checked, including the keyword formatting, the font size in the table, and other formats.

In summary, the author must sort out the logic of the article according to the writing standards, check the details of the article, and make modifications if the content description of the article is unclear. If the author considers continuing to publish the paper in this journal, they should verify and resubmit it.

Minor editing of English language is required.

Author Response

请参阅附件。

Reviewer 2 Report

This manuscript presents an investigation of concrete slabs’ fractal characteristics under low-velocity impacts. Digital image analysis is carried out with a box dimension technique to establish correlation between fractal dimension with energy dissipation. There are some issues to be addressed, especially the novelty.

1. The literature review needs to be done in depth. There are many points in the second paragraph (too long) and readers would get confused. The reviewer suggests splitting it into four aspects: (1) what are the typical characteristics of crack pattern under low-velocity and high-velocity impacts? (2) why not study high-velocity impact? (3) how/why the fractal theory is developed and applied to static and dynamic mechanical problems? (4) what this work have done while the other similar literatures have not, i.e. the novelty needs to be more explicitly presented, as the fractal theory and application are not new.

3. The manuscript reads like a report with routine experiments and softerware usages, but without in-depth analyses or discussions on the dynamic mechanisms. So the authors need to substantially modify the manuscript.

2. Page 3: Figure 1 may be rough. The authors are suggested to draw a 3D diagram to show the geometric dimensions of the test slab and other information on the loading setups.

4. There are typos and editing errors in places. Thorough proofing and improvement on English language are needed.

There are typos and editing errors in places. Thorough proofing and improvement on English language are needed.

Reviewer 3 Report

It is required to unify the format writing of the references

There is limited data that used for analysis the failure characteristics of low-velocity impact concrete slabs

Round 2

Reviewer 2 Report

The authors managed to address the questions and improved the manuscript, which can be accepted for publication.

Need proofing and improvement on English language.